# Deletion of SOCS2 Reduces Post-Colitis Fibrosis via Alteration of the TGFβ Pathway

**DOI:** 10.3390/ijms21093073

**Published:** 2020-04-27

**Authors:** Amna Al-Araimi, Amira Al Kharusi, Asma Bani Oraba, Matar M Al-Maney, Shadia Al Sinawi, Ibrahim Al-Haddabi, Fahad Zadjali

**Affiliations:** 1Department of Biochemistry, College of Medicine and Health Sciences, Sultan Qaboos University, P.O box 35, P.C 123, Muscat 113, Oman; amna.alaraimi3@gmail.com (A.A.-A.); asmaorabah@yahoo.com (A.B.O.); matar@squ.edu.om (M.M.A.-M.); 2Department of Physiology, College of Medicine and Health Sciences, Sultan Qaboos University, Muscat 113, Oman; akharusi@squ.edu.om; 3Department of Pathology, College of Medicine and Health Sciences, Sultan Qaboos University, Muscat 113, Oman; Shadia_72@hotmail.com (S.A.S.); haddabi@squ.edu.om (I.A.-H.)

**Keywords:** inflammatory bowel disease, colitis, growth hormone, suppressor of cytokine signaling protein, fibrosis

## Abstract

Inflammatory bowel disease (IBD) is an immunologically mediated chronic intestinal disorder. Growth hormone (GH) administration enhances mucosal repair and decreases intestinal fibrosis in patients with IBD. In the present study, we investigated the effect of cellular sensitivity to GH via suppressor of cytokine signaling 2 (SOCS2) deletion on colitis and recovery. To induce colitis, wild type and SOCS2 knockout (SOCS2−/−) mice were treated with 3% dextran sodium sulphate (DSS), followed by a recovery period. SOCS2−/− mice showed higher disease activity during colitis with increased mRNA expression of the pro-inflammatory cytokines nitric oxide synthase 2 (NOS2) and interleukin 1 β (IL1-β). At recovery time point, SOCS2−/− showed better recovery with less fibrosis measured by levels of α-SMA and collagen deposition. Protein and mRNA expressions of transforming growth factor beta β1 (TGF-β1) receptors were significantly lower in SOCS2−/− mice compared to wild-type littermates. Using an in vivo bromodeoxyuridine (BrdU) proliferation assay, SOCS2−/− mice showed higher intestinal epithelial proliferation compared to wild-type mice. Our results demonstrated that deletion of the SOCS2 protein results in higher growth hormone sensitivity associated with higher pro-inflammatory signaling; however, it resulted in less tissue damage with less fibrotic lesions and higher epithelial proliferation, which are markers of GH-protective effects in IBD. This suggests a pleiotropic effect of SOCS2 and multiple cellular targets. Further study is required to study role of SOCS2 in regulation of TGFβ-mothers against the decapentaplegic homolog (Smad) pathway.

## 1. Introduction

Inflammatory bowel disease (IBD) describes two conditions: Crohn’s disease and ulcerative colitis, in which chronic inflammation is a hallmark. This results from an inappropriate immune response with a sustained hyperactivity and proliferation of immune cells. An imbalance in cellular signaling, mediated by hormones and inflammatory cytokines, leads to disruptive intestinal homeostasis, mucosal damage and defective repair [1,2].

In a small number of Crohn’s disease patients, administration of growth hormone (GH) showed positive results, with an improved colitis activity index and less need for immunosuppressors [2,3]. Transgenic mice of increased GH expression have better mucosal repair after colitis induction [4] and similar effects were obtained with exogenous administration of GH [5]. Despite the possible benefits from GH therapy in reducing the disease activity via the trophic effects of insulin growth factor-1 (IGF-1), its excess may mediate excessive wound-healing responses that lead to intestinal fibrosis [6,7,8]. A number of clinical studies had reported that patients with chronic colitis have an increased risk to develop colon neoplasia/adenocarcinoma [9,10,11]. Although a general interest in studying the role of growth factors in mucosal repair in models of IBD does exist, none of those studies considered the effect of hypersensitivity of colonic tissues to GH. An alternative to the administration of GH is to increase the cellular sensitivity toward GH, which could be more beneficial to avoid exogenous doses of GH and, therefore, avoiding the reported side effects [12].

Suppressors of cytokine signaling-2 (SOCS2) regulate the intracellular transduction pathway of a GH where it binds to the GH receptor (GHR) and mediates its ubiquitination and further degradation [13,14,15,16]. SOCS2 knockout mice (SOCS2−/−) have increased GH sensitivity and display acromegaly features with higher lean body mass and organ sizes than their wild-type littermates [17,18]. In the intestine, SOCS2 inhibits epithelial cell proliferation [19]. SOCS2 is also shown to exert anti-inflammatory activity and its deletion causes activation of pro-inflammatory signaling pathways and M1 macrophage polarization [13,20]. The effect of SOCS2 on GH sensitivity, inflammation, epithelial repair and growth may suggest its modulation of the disease process in IBD. In this study, we aim to explore the effect of increased GH sensitivity via SOCS2 deletion on the inflammatory process and recovery from colonic inflammation using an established DSS-induced colitis mouse model.

## 2. Results

### 2.1. SOCS2 Deletion Aggravates Colitis Severity in SOCS2−/− Mice

Colitis was induced in mice with 5 days treatment with 3% dextran sodium sulphate (DSS) in drinking water and followed with 12 days recovery (Figure 1A). SOCS2 deletion had been previously shown to have higher pro-inflammatory activity and induced intestinal epithelial proliferation [17,18,19], therefore we included a non-DSS-treated group (pre-DSS) for comparative purposes. During induction of colitis (Days 5–10), SOCS2−/− mice had significant weight reduction compared to the wild type (Figure 1B). Recovery from colitis was observed by an increase in body weight in both mice groups. SOCS2−/− mice showed faster recovery and weight changes were not different at Day 22 of the experiment (Figure 1B). The disease activity index (DAI) score increased after administration of DSS (Figure 1C). SOCS2−/− mice showed higher disease activity during induction of colitis and recovery compared to the wild-type littermates. However, differences in the DAI were statistically not significant at recovery. To further assess differences in disease severity, we performed a microscopic assessment of colitis. Figure 1D shows the average scores for wild-type and SOCS2−/− mice before, during induction and after recovery from colitis. Prior and during the acute phase of colitis, both mice groups showed no difference in the histological score of colitis. However, similar to the DAI, SOCS2−/− mice showed higher colitis severity compared to the wild type at the induction and recovery time point, with *p*-values < 0.01 and <0.001, respectively (Figure 1D). Figure 1E shows sections of the colon during the three phases of the colitis model. At the recovery time point, SOCS2−/− mice had extensive crypt damage, inflammatory cell infiltration and goblet cell depletions. 

### 2.2. Higher Recovery Rate in SOCS2−/− Mice Despite an Increased Inflammatory Process

Cytokines are principal mediators of the innate and adaptive arms of the immune responses in mucosal inflammation; therefore, we investigated the disease activity at a molecular level through the inflammatory cytokines profile in both mice groups during the phase of colitis and recovery. We measured the gene expression of the pro-inflammatory cytokines nitric oxide synthase 2 (NOS2) and interleukin 1 β (IL-1β), as well as the anti-inflammatory interleukin-4 (IL-4) cytokine, in colonic tissues before and after induction of colitis and during recovery. Expression of pro-inflammatory cytokines increased during induction of colitis with no difference observed between the two mice groups. Unlike the wild-type mice, the expression of pro-inflammatory cytokines in SOCS2−/− mice did not decrease during recovery and their levels remained high (Figure 2A). IL-4 gene expressions were not different between the two mice groups. To further support the colonic gene expression, we measured the circulatory levels of cytokines. Plasma levels of interleukin-10 (IL-10) and IL1β could not be measured due to the low detection limit of the assay. Plasma levels of tumor necrosis factor α (TNFα) and IL-4 were not different between the two mice groups (Figure 2B). The data of colonic expression, disease activity and histological scoring (Figure 1) suggest that SOCS2 deletion induced more pro-inflammatory processes compared to the wild-type littermates and disease activity is not explained by changes in anti-inflammatory cytokines. 

SOCS2 deletion is known to increase tissue sensitivity to GH [15,17,18] and GH has earlier shown to have a protective effect in IBD. Thus, we further assessed GH sensitivity and the plasma IGF-1 to GH ratio in both groups during induction and recovery. During induction of colitis and recovery, SOCS2−/− mice showed GH sensitivity compared to the wild-type mice (Figure 2C). This may explain the higher recovery rate in SOCS2−/− mice despite the increased inflammatory process.

### 2.3. SOCS2-Deletion Reduces Fibrosis in an Inflammatory Bowel Model 

Fibrosis is a chronic and progressive pathological process of healing after an inflammatory process and it is characterized by an excessive deposition of extracellular matrix components, such as collagens, leading to scarring of the involved tissue. In IBD, this results in colonic restriction and causes poor quality of life for patients after the inflammatory phase. Administration of GH in rats showed less cecal fibrosis and collagen expression without an effect on intestinal inflammation [21]. We therefore assessed intestinal fibrosis in our model as a consequence of increased GH sensitivity. Figure 3A shows collagen staining in both mice groups in blue color and the color intensity was quantified using ImageJ software (Figure 3B). Less collagen deposition was seen in SOCS2−/− mice at the recovery phase (*p*-value < 0.05). Expression of α-smooth muscle actin (α-SMA), a marker of an active myofibroblast, in the bases of crypts was lower in SOCS2-deleted mice at the three stages (Figure 3C). 

### 2.4. Fibrosis TGFβ1-Smad Signaling Altered by SOCS2-Deletion 

Transforming growth factor β (TGF-β) is a multifunctional polypeptide hormone that influences cellular proliferation, regulation of the inflammatory response, restitution and healing, as well as fibrosis [22]. TGF-β affects virtually all stages of the chronic inflammatory and fibrotic disease processes, at a cellular level [23]. In our study model, we observed decreased expression of TGF-β1 in colonic tissues (significant *p*-value < 0.01 at the pre-DSS stage and <0.05 at the colitis stage) and also plasma levels of TGF-β1 (significant *p*-value < 0.01 at the colitis stage and <0.05 at the recovery stage) in SOCS2−/− mice compared to wild-type littermates (Figure 3D–E). Actions of TGF-β are mediated through the TGF-β receptor (TGF-β R), which is a dimer combination of the three-receptor subunits TGF-β RI, RII and RIII. To further investigate the mechanism behind this observation, we studied the gene expression of TGF-β receptors. We failed to detect expression of TGF-β RI in colonic tissues (data not shown). Expression of TGF-β RII was significantly lower in SOCS2−/− mice prior and after induction of colitis (Figure 3F). TGF-β RIII gene expressions showed high variability between samples (data not shown) and we measured protein expression of TGF-β RIII using Western blotting. TGF-β RIII protein levels were lower in SOCS2−/− mice prior and after induction of colitis (Figure 3G).

### 2.5. Role Epithelial Regeneration in Disease Activity

Additional mechanisms involved in the pathogenesis of IBD include defects in gap junctions and altered colonic epithelial proliferation. We did not see major structural differences in colonic villi, goblet cells nor gap junctions between wild-type mice and SOCS2-knockout mice (Figure 4A). Protein expression of occludin was not different between the two mice groups (Figure 4B). Continuous renewal of the small intestinal epithelium is essential to nutrient digestion and absorption and barrier function. Following intestinal injury and crypt loss, intestinal epithelial cells undergo regeneration, a phenomenon known as adaptive growth or intestinal adaptation. GH administration was shown to enhance intestinal adaptation post radiation damage [24]. To rule out the effect of enhanced GH sensitivity via SOCS2 deletion on intestinal epithelial regeneration, we performed an in vivo bromodeoxyuridine (BrdU) proliferation assay. Colonic tissues from mice were fixed 4 h post intra-peritoneal injection of BrdU. We observed increased intestinal epithelial proliferation in SOCS2-knockout mice compared to the wild-type in anti-BrdU-stained colonic tissues (Figure 4C). Significantly higher BrdU-stained cells per crypt was observed in SOCS2-deleted mice (*p*-value < 0.001).

## 3. Discussion

The pathogenesis of inflammatory bowel disease is mediated by a complex interplay between immune inflammatory and hormonal mediators of intestinal mucosa regeneration and repair. Recent reports described the intestinal recovery effects of GH in mucosal healing and decreased inflammation in experimental colitis [3,5]. Here we demonstrated, via deletion of SOCS2, the effect of increased GH sensitivity on disease pathogenesis of colitis. SOCS2−/− mice showed increased GH sensitivity during the active colitis and recovery periods, and this is associated with increased intestinal epithelial turnover and less collagen deposition at the recovery phase. These effects could be explained by SOCS2 regulation of the TGF-β1 receptor levels in colonic tissues. 

Disease activity during induction of colitis and recovery was higher in SOCS2−/− mice, which was evident during the macroscopic (Figure 1C) and microscopic (Figure 1D) evaluation of disease activity and increased expression of proinflammatory cytokines at the recovery phase (Figure 2A). A high inflammatory state in SOCS2-depleted tissues was earlier documented in a hepatic steatosis model, which showed increased expression of inflammatory cytokines and enhanced NF-kB activation [13]. This was supported by increased expression of pro-inflammatory cytokines in colonic tissues during colitis. The level of anti-inflammatory marker IL-4 was not different from wild-type mice, which is in concordance with earlier findings in liver tissues of high-fat-fed mice [13]. Excessive production of inflammatory cytokines (interleukin-6 (IL-6), iNOS, IL-1β and interferon-γ (IFN-γ)) in SOCS2-deleted mice could be mediated via its actions on macrophage activation and alteration in pro-inflammatory signaling cascades [13]. Earlier, we described a new mechanism of SOCS2-mediated regulation of inflammatory pathways, through serine–threonine kinase non-race-specific disease resistance-1 (NDR1) and TNFα-mediated nuclear factor kappa-light-chain-enhancer of activated B (NF-kB) activation [20]. SOCS2 directly binds to NDR1 and mediates its ubiquitination [20,25,26]. Deletion of SOCS2 execrates the inflammatory pathways, which increase disease activity during colitis. 

Inflammatory processes induce healing processes and regenerative fibrosis that may progress in a self-perpetuating manner and ultimately lead to organ scarring and subsequent loss of function [23]. Growth hormone activity was earlier described as having a beneficial effect on IBD in human and animal models, specifically in its anti-fibrotic activity [2,3,4,6,7,21]. SOCS2 is a pleotropic intracellular protein that target tyrosine phosphorylated targets via its SH2 domain [15,16,25,26,27]. It may target cellular pathways other than the GH–signal transducer and activator of transcription 5 (STAT5) pathway and explain the recovery observed in our model. Non-immune cell types (e.g., mesenchymal cells) are involved in IBD fibrotic progression [28,29,30] through the regulation of growth factors to produce excessive amounts of extracellular matrix (ECM) proteins during fibrosis [31,32,33]. Myofibroblasts are the predominant mucosal cells that synthesize components of the ECM [34]. Its activation increases in abnormal healing processes, which may divert the healing process towards fibrogenesis. Using α-SMA, a marker of a differentiated myofibroblast, we observed a decrease in SOCS2−/− compared to wild-type mice. This was supported with less collagen deposition and TGF-β1 expression. This could also be mediated by the anti-fibrotic activity of interferon gamma [35], which we earlier showed to be highly elevated in SOCS2-deleted mice compared to the wild-type [13]. TGF-β1 is a key mediator in the pathogenesis of fibrosis and it mediates progressive fibrosis by stimulating ECM production while inhibiting its degradation. Active TGF-β binds to its receptors and functions as autocrine and paracrine manners to exert its biological and pathological activities via mothers against decapentaplegic homolog (Smad)-dependent and independent signaling pathways [36]. The co-receptor TGF-β1RIII and the receptor TGF-β1RII are key mediators of TGF-β signaling. They belong to the serine and threonine kinase receptors family like GH receptor [37]. SOCS2 binds to the GH receptor and mediates it ubiquitination and degradation [15,25]. Negative regulation of TGF-β1 receptors involve binding to Smad7 that further recruits multiple E3-ligases, such as Smad ubiquitin regulatory factors (Smurf1 and Smurf2), neural precursor cells expressed developmentally downregulated proteins (NEDD4 and 2) and WW Domain Containing E3 Ubiquitin Protein Ligase 1 (WWP1), to the receptor, leading to ubiquitination and degradation of the receptor [38,39,40,41]. The effect of SOCS2 deletion in reducing TGF-β1 receptor levels could be mediated via alteration of the E3-ligases that target the receptor. More investigations are warranted to study the role of SOCS2 in regulation of the TGF-β 1/Smad signaling pathway. 

We also observed increased epithelial proliferation in SOCS2-knockout mice. Exogenous GH administration in a sepsis-induced colitis rat model showed GH induced IGF-1 expression in the colon. This prevented intestinal atrophy in septic rats, as well as protected the integrity of the intestinal structure and mucosa barrier [42]. Further support to the protective role of GH came from necrotizing enterocolitis and radiotherapy-induced colitis models [24,43]. This effect is mediated via activated STAT5 signaling [44] that earlier was shown to be prolonged after SOCS2 deletion [15]. Increased adaptive growth could account for less fibrosis in SOCS2-deleted mice despite a higher disease activity. 

In conclusion, our results demonstrate that deletion of the SOCS2 protein causes exacerbated colonic inflammation during colitis but is associated with decreased fibrosis at recovery measured by levels of α-SMA and collagen deposition. This is explained via a mechanism that involves alteration of the TGF-β 1/Smad signaling pathway and increased intestinal epithelial cell proliferation. This study also provides a new mechanistic understanding of GH’s role and interaction with the SOCS2 protein in the disease activity of IBD through fibrotic pathways. Further study is warranted to investigate the cellular and molecular mechanisms of the SOCS2-mediated fibrotic pathways. These findings provide key data to further test administration of GH in reducing fibrotic lesions in recurrent colitis. 

## 4. Materials and Methods 

### 4.1. Mice and Induction of Colitis and Recovery

Male wild-type (SOCS2+/+) and SOCS2-KO (SOCS2−/−) C57BL/6J mice were used, aged between 12 and 16 weeks. Mice were housed under standard conditions (12:12 h light–dark cycles, 22 ± 2 °C and about 60% humidity) with ad libitum feeding. A total of 10 mice per group was used. Mice were allowed to acclimate to the cage for a week before experiments. Dextran sodium sulphate is a synthetic sulfated polysaccharide that induce colonic inflammation that resembles ulcerative colitis. Acute colitis was induced in mice using 3.0% (*w*/*v*) dextran sodium sulphate (DSS; MW 36,000–50,000; T&D consulting, Uppsala, Sweden) in drinking water for 5 days. After this, mice were switched to normal water and monitored for additional 12 days to observe recovery from colitis (Figure 1A). The disease activity index (DAI) was measured by percentage change of basal body weight, stool consistency and rectal bleeding. The DAI was earlier adopted and showed good histological correlation [45,46,47]. The percentage of weight-loss scale was as follows: 0: 1%, 1: 0%–4.99%, 2: 5.00%–9.99%, 3: 10.00%–19.99% and 4 ≥ 20.00%. Fecal bleeding score: 0 = none, 3 = slightly positive, 5 = positive, 6 = strongly positive using hemoccult test (hemo FEC kit; Cobas, India); the stool consistency score was as follows: 0 = well-formed pellets, 2 = form pellet but loose stool and 4 = liquid stool or no formed pellet [45]. The average of the sum of the three scores gave the DAI.

Mice were sacrificed after anesthesia by isoflurane by total bleeding from the orbital plexus followed by cervical dislocation. Mice were sacrificed at three time points: pre-DSS (at day 5), after induction of colitis (at day 10) and recovery (at day 22) (Figure 1A). The entire colon was dissected, and contents were flushed with ice-cold saline. The distal third of the colon was dissected and split into three (2–5 mm) segments for histological analysis and RNA and protein extractions. All animal protocols in this study were approved by the Animal Ethics Committee of Sultan Qaboos University (SQU/AEC/2011-12/2). 

### 4.2. Histological Evaluation of DSS-Induced Colitis Severity 

Paraffin-embedded colonic sections were stained using the hematoxylin–eosin (H&E) method. Briefly, sections were rehydrated as following: dewax in Xylene for 3 min (x2), 100% ethanol for 3 min (x2), 90% ethanol for 3 min (x2) and 70% ethanol for 3 min (x2), followed by a wash with running water for 3 min. Sections were then stained in Harris Hematoxylin (American MasterTech Scientific Ltd., CA, USA) for 10 min, washed in running water for 3 min, followed with one dip in 1% acid/alcohol and immediately in running water till blue color development. Sections were then counter stained in 1% eosin (J.T.Baker, USA or TCS Biosciences Ltd., UK) for 5 min followed with washing in running water. Sections were dehydrated as follows: 70% ethanol for 30 s, 90% ethanol for 30 s, 100% ethanol for 1 min (x2) and xylene for 3 min (x2). Sections were then mounted with a coverslip with DPX mountant (Surechem Products Ltd., Ipswich, UK). Stained sections were then observed under same light illumination and images were taken at different magnifications using an Olympus microscope with an attached Olympus digital camera (Olympus, Tokyo, Japan). The specimens were evaluated for histological signs of colonic inflammation in a double-blind fashion by two independent pathologists; the average score for each mouse was then determined. The scoring of histological changes in the colon was done according to Cooper et al. [46], which reflects the extent and severity of colon inflammation, crypt loss, infiltration of neutrophils, lymphocytes and goblet cell depletion (Table 1). Tissues were also stained with Masson’s trichrome for collagen deposition. 

### 4.3. Immunohistochemistry

Sections of colon (3 μm thick) were dewaxed in xylene, hydrated through descending concentrations of ethanol and equilibrated with PBS, followed by antigen retrieval in a preheated citrate buffer at 95 °C, pH 6.0 for 30 min. Endogenous peroxidase activity was quenched by 3% H_2_O_2_, and unspecific sites were blocked by horse serum in PBS prepared according to the manufacturer’s instructions (VECTASTAIN^®^ Universal ABC Kit; Vector Laboratories, Burlingame, CA, USA). Then, the sections were incubated overnight at 4 °C with a primary of a mouse monoclonal antibody against α-SMA (Sigma-Aldrich, St. Louis, MO) at a dilution of 1:550 in normal blocking serum or anti-bromodeoxyuridine (Clone MoBU-1, Thermo Fisher Scientific, Waltham, MA, USA) at a dilution of 1:50, followed by incubation with a biotinylated secondary antibody (anti-Mouse IgG/Rabbit IgG) and streptavidin-HRP conjugate (VECTASTAIN^®^ ABC Reagent, , Burlingame, CA, USA). Thereafter, the sections were incubated with a chromogen DAB (Invitrogen, Boston, MA, USA) and stained with hematoxylin. Images were captured at the same magnification and illumination. Negative controls of the no BrdU-treated mice were run in parallel in all experiments.

### 4.4. In Vivo Epithelial Cell Proliferation 

To evaluate cell proliferation in mice groups, bromodeoxyuridine (BrdU, Sigma-Aldrich; B5002) was dissolved in PBS at 10 mg/mL. Mice (*n* = 5 per group) were injected with the BrdU solution intraperitoneally at 50 mg/kg body weight, then sacrificed after four hours. The distal colons of the mice groups were resected and fixed overnight with 4% paraformaldehyde and embedded in paraffin. Six-micrometer sections were cut and incubated with anti-BrdU (Clone MoBU-1, Thermo Fisher Scientific, Waltham, MA, USA) as a primary antibody and goat anti-mouse HRP conjugated antibody (Santa Cruz Biotechnology) as a secondary antibody. Following immunostaining for BrdU, the number of BrdU-positive cells was counted from 20–23 well-oriented crypts per mouse per genotype. 

### 4.5. Quantitation of Cytokine Gene Expression Using Real-Time PCR 

To further assess disease activity, we measured the gene expression of different markers of inflammation or fibrosis in colonic tissues using SYBR^®^ Green quantitative PCR. Pro-inflammatory cytokines (*IL1β* and *NOS2*), anti-inflammatory cytokines (*IL-4*) and fibrotic markers (TGFβI, TGFβRI and TGFβRII) were measured. Colonic RNA was extracted from frozen colonic tissue using TRIzol reagent (Invitrogen, New Zealand). RNA purity and quantification were determined using a NanoDrop spectrophotometer (Nanodrop Technologies). Reverse transcription reactions were performed with iScript™ cDNA Synthesis Kits (Bio-Rad, Hercules, CA, USA). SYBR green real-time PCR was performed on the ABI 7500 Fast Real Time PCR System instrument (Applied Biosystems, Austin, TX, USA) by following the iQ^TM^ SYBGR^®^ Green Supermix manual (Bio-Rad, Hercules, CA, USA). Results were reported as the relative expression of the genes corrected for expression of the *β-Actin* gene. Primer sequences are listed in Table 2. 

### 4.6. Plasma Levels of Cytokines 

To quantify the plasma cytokine levels of inflammatory and anti-inflammatory cytokines, we used the Milliplex ELISA kit for IL-1β, TNF-α, IL-4, IL-10 and GH. Murine cytokine TGF-β1 (R&D System, Minneapolis, MN, USA) and IGF-1 (Crystal Chem Inc., Downers Grove, IL, USA) plasma levels were also measured by colorimetric ELISA kits according to the manufacturers’ protocol. 

### 4.7. Western Blot Analysis

Colonic tissues were lysed in lysis buffer containing protease inhibitors and phosphatase inhibitors and extracted proteins were separated in SDS-PAGE gels and transferred to polyvinylidenediflouride (PVDF) membranes (Millipore). After blotting, membranes were blocked in 5% non-fat dry milk or BSA (Sigma) in Tris-Buffered Saline (TBS) containing 0.1% Tween 20. Membranes were incubated with the following primary antibodies as specified: anti-TGFβRIII (Cell Signaling), Occludin (Invitrogen) and anti-GAPDH (Calbiochem). HRP-conjugated secondary antibody (Santa Cruz) was used. Membranes were visualized with the ECL Western blotting detection system (Luminata Crescendo Western HRP Substrate; Millipore) according to the manufacturer’s instruction. 

### 4.8. Transmission Electron Microscopy Analysis

Ultra-structural changes of the colon, such as the gap junction, villi and goblet cells, were examined using transmission electron microscopy. The colon was dissected as ≤0.5 cm in length, followed by fixation with a fixative buffer (paraformaldehyde, 25% glutaraldehyde, 1 N NaOH, 1 N HCl, sucrose, 1% CaCl, 1% MgCl and dH_2_O). Blocks were trimmed and sectioned into semi-thin (0.5 µm thick) and then for ultra-thin (60–90 nm thick). The sections were placed in grids and stained with uranyl acetate (R1260 A, Agar) for 30 min and washed first with 50% alcohol then with free CO_2_-distilled water. To ensure the removal of CO_2_ from the sections, sodium hydroxide was used during the staining step. The sections were then stained with lead citrate for 30 min following washing with free CO_2_-distilled water. Finally, sections were examined under TEM and images were captured for analysis.

### 4.9. Statistical Analysis

Data were expressed as the mean ± SEM. GraphPad Prism software was employed to analyze the data. Significant differences between the two groups of mice were assessed using Student’s *t*-tests. All *p*-values less than 0.05 were considered significant.

## Figures and Tables

**Figure 1 ijms-21-03073-f001:**
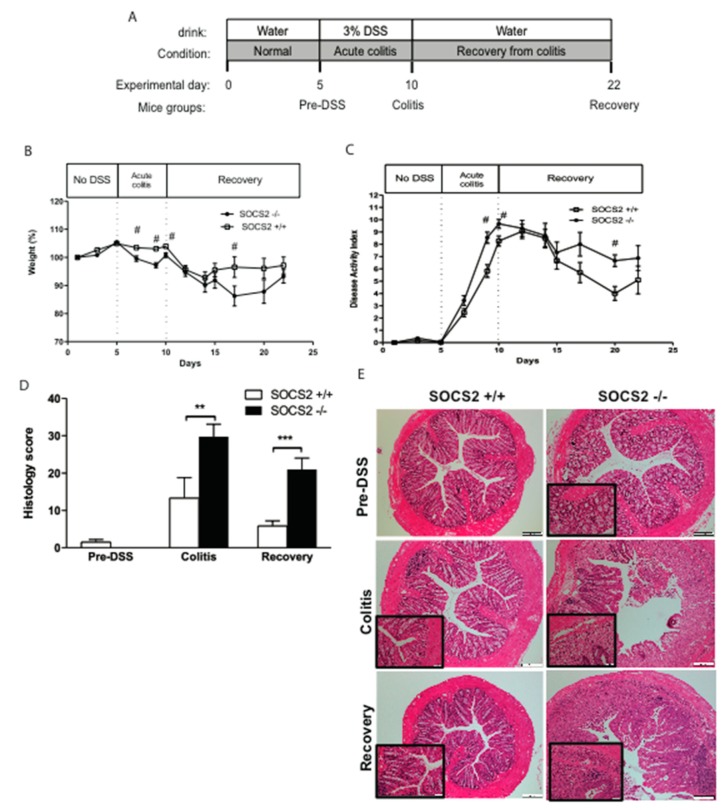
SOCS2 deletion aggravates colitis severity in mice. (**A**) Experimental design of dextran sodium sulphate (DSS)-induced colitis. Both wild-type (SOCS2+/+) and SOCS2-knockout (SOCS2−/−) mice were sacrificed at three time points. Prior colitis (Pre-DSS), after DSS-induced colitis and at recovery 12 days after abstaining DSS from drinking water. Percentage of body weight changes (**B**), macroscopic colitis severity index (**C**) and histological colitis score (**D**) in SOCS2+/+ and SOCS2−/− mice at three time points of the model. (**E**) Hematoxylin and eosin staining of the distal colon from mice at the three time points of the model taken at 10x magnification. Magnified sections (X40) are shown in the bottom left corner. Details of the macroscopic and histological assessments are described in the Methods section. Student’s *t*-test *p*-values: # <0.05, ** <0.01, *** <0.001; *n* = 10 mice per group.

**Figure 2 ijms-21-03073-f002:**
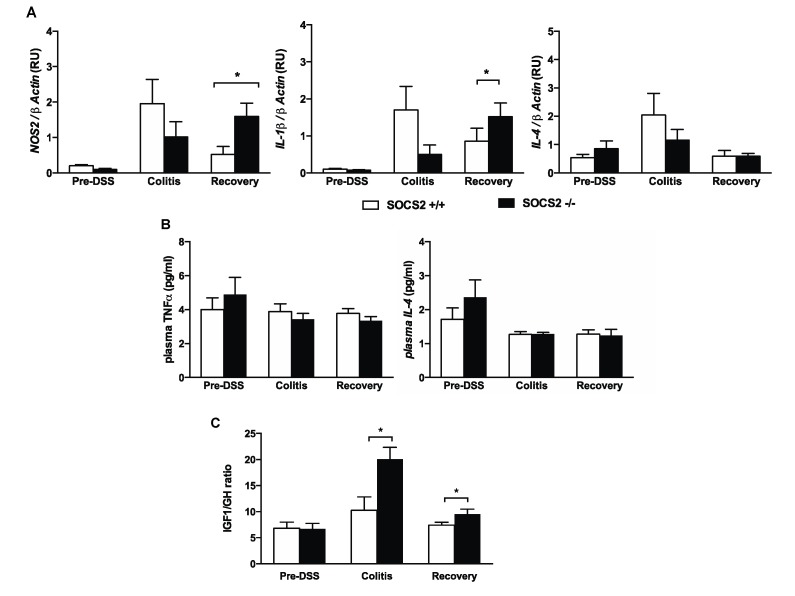
Growth hormone and inflammatory activity during colitis and recovery. (**A**) Gene expressions of pro-inflammatory (NOS2 and IL-1β) and anti-inflammatory cytokines (IL-4) in colonic tissue prior, during DSS-induced colitis and at recovery in wild-type (SOCS2+/+) and SOCS-knockout (SOCS2−/−) mice. Gene expressions were measured using the relative standard method and normalised to expression of the β-actin gene. Data are shown in relative unit (RU). (**B**) Plasma cytokine levels of TNFα and plasma IL-4 of in both mice groups. (**C**) Growth hormone (GH) sensitivity index measured as the ratio of plasma IGF-1/GH in both mice groups during the three time points of the model. * Student’s *t*-test *p*-value < 0.05; *n* = 6 per mice group for the gene expression study and for plasma analysis *n* = 10 per group.

**Figure 3 ijms-21-03073-f003:**
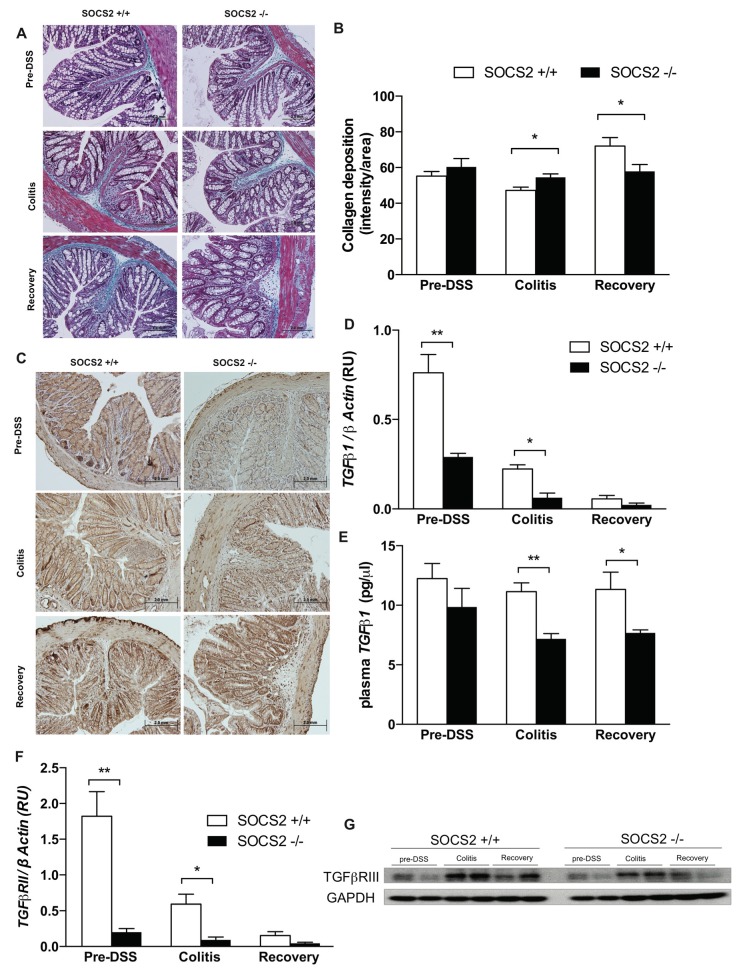
SOCS2-deletion induces less fibrosis in an inflammatory bowel model. (**A**) Masson’s Trichrome staining taken at 20X magnification prior, at colitis and after recovery from colitis in wild-type (SOCS2+/+) and SOCS-knockout (SOCS2−/−) mice. (**B**) Collagen staining intensity in the study model measured from color threshold analysis using ImageJ tool. An average of 3 sections from each mouse was taken in a group of 5 mice. (**C**) immunohistochemistry of α-SMA in colonic sections. (**D**,**E**) TGFβ1 colonic gene expressions and plasma levels in both mice groups at the three time points of the study: pre-colitis, colitis and recovery. (**F**) TGFβRII gene expressions. (**G**) Western blot of TGFβ-receptor III levels in both mice groups at the three time points of the disease model (*n* = 2 per group). All gene expressions were normalized to the expression of the β-actin gene. Student’s *t*-test *p*-values: * < 0.05, ** < 0.01.

**Figure 4 ijms-21-03073-f004:**
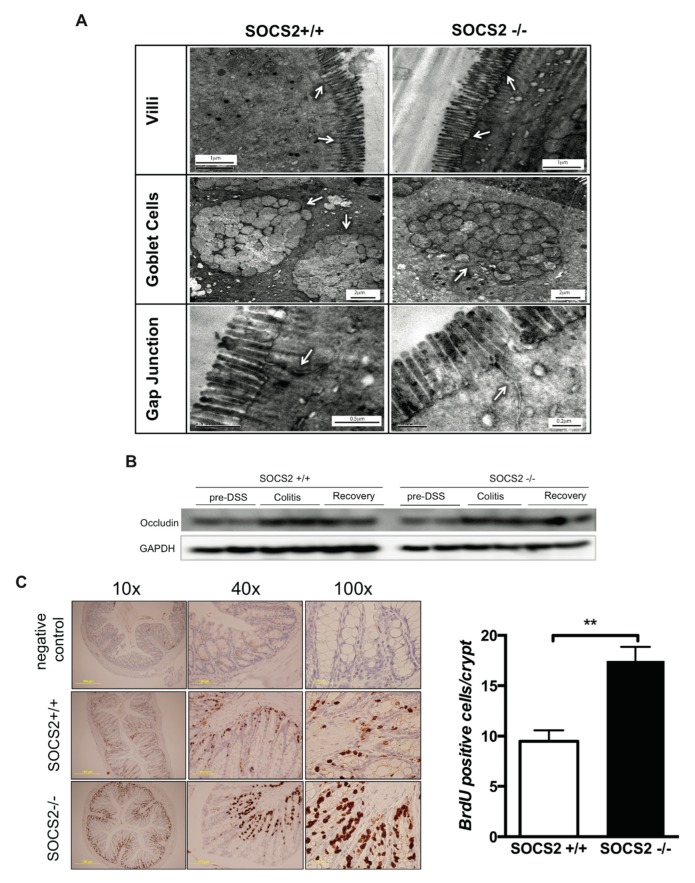
Colonic microscopic changes and epithelial regeneration. (**A**) Transmission electron microscope images of colons of both mice not treated by dextran sodium sulphate (DSS). The ultra-structure of the villi, goblet cells and gap junctions (white arrows) are shown. (**B**) Western blot of the gap junction protein occludin. (**C**) In vivo colonic epithelial proliferation measured by immunohistochemical staining of BrdU in colonic tissues of mice without treatment with DSS. Colonic tissues were fixed 4 h post intraperitoneal injection of BrdU. Sections are shown in 10× and 40× magnification. Negative control tissues are from mice not injected with BrdU. Total positive cells per crypts are shown in the bar diagram. A total of 20–23 crypts were studied per mouse sample and 5 mice per group were studied. Student’s *t*-test *p*-value ** < 0.01.

**Table 1 ijms-21-03073-t001:** Histological scoring of the severity of the colitis.

Feature	Score	Description
Inflammation severity	0	None
	1	Mild
	2	Moderate
	3	Severe
Inflammation extent	0	None
	1	Mucosa
	2	Submucosa
	3	Transmural
Crypt damage	0	None
	1	Basal 1⁄3 damaged
	2	Basal 2⁄3 damaged
	3	Crypt lost
	4	Surface epithelial lost
Ulcer	4	
Lymphocyte infiltration	3	
Neutrophil infiltration	2	
Cryptitis	3	
Crypt abscess	3	
Edema	4	
Goblet cell depletion	3	

**Table 2 ijms-21-03073-t002:** List of primers and sequences used for quantitative PCR.

Gene	Forward Primer	Reverse Primer
*NOS2*	CATCAACCAGTATTATGGCTC	TTTCCTTTGTTACAGCTTCC
*IL-1β*	GGATGATGATGATAACCTGC	CATGGAGAATATCACTTGTTGG
*IL-4*	CTGGATTCATCGATAAGCTG	TTTGCATGATGCTCTTTAGG
*TGFβ1*	CACCGGAGAGCCCTGGATA	TGTACAGCTGCCGCACACA
*TGFβ* *RI*	TGCAATCAGGACCACTGCAATAA	GTGCAATGCAGACGAAGCAGA
*TGFβ* *RII*	AAATTCCCAGCTTCTGGCTCAAC	TGTGCTGTGAGACGGGCTTC
*β* *-Actin*	GATGTATGAAGGCTTTGGTC	TGTGCACTTTTATTGGTCTC

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
