# Peer review of "Deletion of SOCS2 Reduces Post-Colitis Fibrosis via Alteration of the TGFβ Pathway"

_ijms, 2020, doi:10.3390/ijms21093073_

Round 1

Reviewer 1 Report

The manuscript Al-Araimi et al. describes the role of SOCS2 (suppressor of cytokine signalling-2) to promote inflammation and fibrosis in murine inflammatory disease (IBD), desensitizing the protective effect of growth hormone (GH).

The authors convincingly demonstrate that SOCS2 may be the one of mechanistic key responsible to the protective effects shown by GH in IBD patients. Overall, this is a well performed and comprehensive study. There are a few points the authors should address to further strengthen their study.

Major points

-The authors decided to evaluate some pro-, anti-inflammatory cytokines and fibrotic genes with RT-PCR and ELISA. The cytokines evaluated do not described completely the model. As pro-inflammatory cytokines they evaluated Nos2 and Il1b in colonic tissues, and Il4 as an anti-inflammatory one. They also evaluated TNF, IL-4 in mice serum, and IL-1b and IL-10 without detecting anything. Omitting the sensibility of ELISA assays, I think it would be complete to evaluate other genes in colonic tissues. For example, TNF, correlating to serum values, but also IL-10. Other pro-inflammatory genes associated to fibrosis could be Cxcl10 or Il36 or IL17A

-Lane 61: The authors said they used 5% of DSS during the in vivo experiments. While they described they used 3% DSS in the Abstract and in Materials & Methods section. Write the right concentration.

-In Figure 1D, they evaluated the histological score in the three different time points of DSS experiment. I think it is better to underline the days in which they harvested the tissues and why they chose that day. For example, observing the weight curve, at day 14 of the recovery period, both WT and SOCS2-/- showed a more evident body weight loss. Choosing the tissues from this day wouldn’t have shown the same results in the histological score. Explain the choice.

Minor points

There are several misspellings need corrections:

-Lane 21: mRNA

-Lane 22: in vivo and not In-vivo

-Lane 52: intestine and not Intestine

-Lane 61: dextran

-Lane 99: eliminate double is

-Lane 117: it is characterized

-Lane 124: deleted

-Lane 125: three

-Lane 177: describes

Author Response

We are thankful for the reviewer comments. Kindly please find our replies to the comments below:

Comment #1:

-The authors decided to evaluate some pro-, anti-inflammatory cytokines and fibrotic genes with RT-PCR and ELISA. The cytokines evaluated do not described completely the model. As pro-inflammatory cytokines they evaluated Nos2 and Il1b in colonic tissues, and Il4 as an anti-inflammatory one. They also evaluated TNF, IL-4 in mice serum, and IL-1b and IL-10 without detecting anything. Omitting the sensibility of ELISA assays, I think it would be complete to evaluate other genes in colonic tissues. For example, TNF, correlating to serum values, but also IL-10. Other pro-inflammatory genes associated to fibrosis could be Cxcl10 or Il36 or IL17A

reply:

The list of cytokines for inflammation, pro-inflammation and fibrosis are many. In our project we used milliplex to measure serum cytokines. Due to budget restrictions we selected few of these markers. Our gene expression analysis for TNF-a and IL-10 did not show reliable data due to large variation with the group. For fibrotic markers, we also ran procollagen1 gene expression and it resulted in large variation within the group. It is difficult to run gene expression with new set of primers as current COVID-19 crisis we don’t get easy access to reagents shipping and primers.

In the current manuscript, we have convincing evidence on inflammation and fibrosis demonstrated by disease activity index, histological scoring, immunohistochemistry gene expression and also plasma levels of some of the cytokines. Therefore, we believe these results are sufficient to justify the findings of the model.

Comment#2   ‘’Lane 61: The authors said they used 5% of DSS during the in vivo experiments. While they described they used 3% DSS in the Abstract and in Materials & Methods section. Write the right concentration.’’

Reply: the statement is corrected now , we used 3% DSS in the model.

Comment #3  ‘’In Figure 1D, they evaluated the histological score in the three different time points of DSS experiment. I think it is better to underline the days in which they harvested the tissues and why they chose that day. For example, observing the weight curve, at day 14 of the recovery period, both WT and SOCS2-/- showed a more evident body weight loss. Choosing the tissues from this day wouldn’t have shown the same results in the histological score. Explain the choice.

Reply:

We described in the method section the date of sacrifice, please see line#283 “Mice were sacrificed at three time points: pre-DSS (at day 5), after induction of colitis (at day 10) and recovery (at day 22)’’.

At the time of conducting the experiment, we were interested in having  a recovery stage to analyze the effect of increased GH signaling. We defined it by a stage where mice started to stabilize their body weight and regain it. Figure 1B shows that form day 15 the weight start to raise after they were decreasing as an effect of DSS. The decision of sacrifice at day 22 was made when the body weight were around 95%.

Minor points

There are several misspellings need corrections:

-Lane 21: mRNA

-Lane 22: in vivo and not In-vivo

-Lane 52: intestine and not Intestine

-Lane 61: dextran

-Lane 99: eliminate double is

-Lane 117: it is characterized

-Lane 124: deleted

-Lane 125: three

-Lane 177: describes

reply:  all comments are corrected now

Reviewer 2 Report

Review of the manuscript “Deletion of SOCS2 reduces post-colitis fibrosis via alteration of TGFβ pathway” by Amna Al-Araimi et all submitted to “International Journal of Molecular Sciences”.

The authors studied the effects of increased GH sensitivity via SOCS2 deletion on the inflammation and recovery from DSS-induced colitis. The topic is important and the manuscript will be interesting for the readers of “International Journal of Molecular Sciences”

Although the submitted article is well prepared I recommend to introduced the following corrections prior to publication.

  • Due to using a lot of abbreviations, I the authors should check if the abbreviation is preceded by the full name, e.g. in the abstract: SOCS2, NOS2, IL1b, TGFb,….
  • The introduction should provide sufficient background on the topic. I miss the information about IBD diseases (main two Crohn's disease and ulcerative colitis) and DSS-induced colitis model. Both diseases are included in IBD, but there differ significantly. Is DSS-induced colitis is a model of Crohn's disease or ulcerative colitis?
  • I have some doubts with regard to experimental design. On what basis the authors have chosen different experimental time points? Did the authors perform pilot stud? Why authors selected Pre-DSS on 5 day of the experiment (in my opinion it is in fact 0 day of the experiment). The health status of the animals on day 5 was the same as after acclimatization. In the ideal experimental setup in each time point (in this case two: acute colitis and recovery) authors should have health and with DSS- induced colitis groups of both types of mice (SOCS2-/-, SOCS2+/+). Actually, at this setup authors have not had corresponding control at each time point.
  • In section 2.2-2.5, although authors performed statistical analysis for inflammatory markers, some of their arguments were confusing and not based on the results of the analysis. For example, in line 98, they described “The colonic expression data suggest that SOCS2 deletion induced more pro-inflammatory processes compared to wild-type littermates...”. However, NOS2, IL-1b relative expression were significantly up-regulated only in animals during recovery, but not acute inflammatory period. In the section 2.2-2.5 authors, only one time referenced to the p-value of statistical analysis. The authors should read their manuscript and rewrite statistical results more carefully, especially in section 2.3~2.5.
  • Probably it will be also more informative to analyses the level of proinflammatory cytokines in colon tissue than in plasma. If it will be impossible authors should discuss this topic based on the literature.
  • Line 183 – 184 “Disease activity during induction of colitis and recovery was higher in SOCS2−/− mice, which was evident histologically with increased expression of proinflammatory cytokines.” The author's statement is not supported by results. This is true only in the recovery phase.
  • Line 201-204 The sentence is not finished. Non immune cell types (e.g. mesenchymal cells) are involved in IBD fibrotic progression [28-30] though regulation of growth factors to produce excessive amounts of extracellular matrix (ECM) proteins during [31-33].”
  • The conclusion should be improved based on the significant results.
  • I recommend combining the section Mice with Induction of colitis and recovery into one section – experimental design. The descriptions are replicated.
  • Colon fixation for H&E staining should be described.
  • How many mice were used to BrdU – this information should be provided in method description.
  • Line 291-292 “…gene expression of different cytokines of inflammation or fibrosis in colonic tissues using…” it should be: “…gene expression of different markers of inflammation or fibrosis in colonic tissues using…
  • Two-way ANOVA followed by post-hoc test should be performed to assess the effects between different time-points with two main factors: condition (health, acute colitis, recovery from colitis) and mice litter (SOCS2 wild type and with deletion).

Author Response

replies are in the attached pdf file

Reviewer 3 Report

The authors study the effect of GH administration on an experimentally induced IBD model. In particular It has been investigated the effect of cellular sensitivity to  GH via SOCS2 deletion on colitis and recovery. To achieve this, authors combine classical histological techniques of light microscopy (including immunohistochemistry) and electron microscopy with that of biochemistry and molecular biology. The results demonstrated that deletion of SOCS2 protein results is related to higher growth hormone sensitivity in turn associated with higher pro-inflammatory signaling with less fibrotic lesions and higher epithelial proliferation, which are markers of GH-protective effects in IBD. This suggests pleiotropic effect of SOCS2 and multiple cellular targets. The manuscript answers all scientific curiosities through numerous techniques explained in detail. As for understanding, while imposing an effort in understanding the project (and therefore the reader if not expert in these matters is helped by a rich bibliography) the manuscript flows very well in reading. It follows that I strongly suggest publication in this journal

Author Response

We are thankful and much grateful for the reviewer positive feedback. You have made our day brightful  with the feedback.  

Round 2

Reviewer 1 Report

The authors have answered at all the recommendations and I think that the manuscript is suitable for publication.

Reviewer 2 Report

Since the authors improved the manuscript the study should be accepted for publication.